# Transfer Q-Learning with Composite MDP Structures

Jinhang Chai [1]  Elynn Chen [2]  Lin Yang [3]

## Abstract

To bridge the gap between empirical success and theoretical understanding in transfer reinforcement learning (RL), we study a principled approach with provable performance guarantees. We introduce a novel composite MDP framework where high-dimensional transition dynamics are modeled as the sum of a low-rank component representing shared structure and a sparse component capturing task-specific variations. This relaxes the common assumption of purely low-rank transition models, allowing for more realistic scenarios where tasks share core dynamics but maintain individual variations. We introduce UCB-TQL (Upper Confidence Bound Transfer Q-Learning), designed for transfer RL scenarios where multiple tasks share core linear MDP dynamics but diverge along sparse dimensions. When applying UCB-TQL to a target task after training on a source task with sufficient trajectories, we achieve a regret bound of $\widetilde{\mathcal{O}}(\sqrt{eH^5N})$ that scales independently of the ambient dimension. Here, $N$ represents the number of trajectories in the target task, while $e$ quantifies the sparse differences between tasks. This result demonstrates substantial improvement over single task RL by effectively leveraging their structural similarities. Our theoretical analysis provides rigorous guarantees for how UCB-TQL simultaneously exploits shared dynamics while adapting to task-specific variations.

## 1. Introduction

Transfer reinforcement learning (RL) has emerged as a promising solution to the fundamental challenge of sample inefficiency in RL. By leveraging knowledge from related tasks, transfer learning aims to accelerate policy learning and improve performance in new environments without requiring extensive data collection. This approach has shown empirical success across various domains, from robotics to game playing, yet theoretical understanding of how transfer provably benefits RL remains limited.

Consider autonomous vehicle training as an illustrative example: core driving dynamics – including vehicle physics, road rules, and basic navigation – remain consistent across different driving scenarios. However, specific environments (urban vs. highway driving, varying weather conditions, different traffic patterns) introduce distinct variations to these core dynamics. This naturally suggests modeling transition dynamics as a combination of shared low-rank structure capturing common elements, plus sparse components representing scenario-specific variations.

We propose a composite MDP framework that formalizes this intuition: transition dynamics are modeled as the sum of a low-rank component representing shared structure and a sparse component capturing task-specific deviations. This structure appears in many real-world applications beyond autonomous driving – robotic manipulation with different objects, game playing across varying environments, and resource management under changing constraints all exhibit similar patterns of *core shared dynamics* with *sparse task-specific variations*.

Our approach extends existing work in several important directions. Prior transfer and multi-task RL research has primarily focused on pure low-rank MDPs (Agarwal et al., 2023; Lu et al., 2021; Cheng et al., 2022) or made direct assumptions about value or reward function similarity (Calandriello et al., 2014; Du et al., 2024; Chen et al., 2024; Chai et al., 2025). While sparsity has been studied in the context of value function coefficients, theoretical analysis of sparse transition structures – particularly in combination with low-rank components – remains unexplored. This gap is significant because transition dynamics often more directly capture task similarity than value functions.

We begin by addressing single-task learning within this composite structure, introducing a variant of UCB-Q-learning tailored specifically for composite MDPs, which may involve a high-dimensional ambient space. In contrast to

---

[1]Department of Operations Research and Financial Engineering, Princeton University [2]Department of Technology, Operations, and Statistics, New York University [3]Department of Electrical and Computer Engineering, UCLA. Correspondence to: Elynn Chen <elynn.chen@stern.nyu.edu>, Lin Yang <linyang@ee.ucla.edu>.

*Proceedings of the 42nd International Conference on Machine Learning*, Vancouver, Canada. PMLR 267, 2025. Copyright 2025 by the author(s).

previous work, we consider the high-dimensional setting where the feature dimensions $p, q \gg$ number of trajectories $N$, and the transition core $M^*$ is no longer a low-rank matrix. This departure from low-rank structures makes existing algorithms designed for linear MDPs inapplicable. Similarly, methods built for low-rank MDPs fail in our context due to the absence of low-rank assumptions in $M^*$.

Our work provides the first theoretical guarantees for this setting, demonstrating how the algorithm successfully learns both shared and task-specific components. These results extend and complement the existing body of work on low-rank MDPs by explicitly handling structured deviations from low-rank assumptions (Du et al., 2019b; Lattimore et al., 2020). Unlike the approach in (Foster et al., 2021), which introduced a Decision-Estimation Coefficient (DEC) to characterize the statistical complexity of decision-making across various scenarios, our framework relies on distinct structural assumptions. This necessitates the development of new techniques, as discussed in detail in Section 3.3.

Building on this foundation, we propose UCB-TQL (Upper Confidence Bound Transfer Q-Learning) for transfer learning in composite MDPs. UCB-TQL strategically exploits shared dynamics while efficiently adapting to task-specific variations. Our theoretical analysis demonstrates that UCB-TQL achieves dimension-independent regret bounds that explicitly capture dependencies on both rank and sparsity, showing how structural similarities enable efficient knowledge transfer. In particular, we construct a novel confidence region (CR) for the sparse difference, thereby reducing the target sample complexity in the online learning process, as discussed in detail in Section 4.3.

Our primary contributions are as follows.

- A novel composite MDP model that combines *low-rank shared structure* with *sparse task-specific components*, while allowing high-dimensional feature spaces. This framework better captures real-world task relationships and provides a foundation for future work in multi-task and meta-learning settings.

- The first theoretical guarantees for single-task RL under the high-dimensional composite transition structure, demonstrating how algorithms can effectively learn and utilize both shared and task-specific components.

- A transfer Q-learning algorithm with *provable regret bounds* that explicitly characterize how structural similarities enable efficient knowledge transfer across tasks.

This work represents a significant step toward bridging the gap between empirical success of transfer RL and theoretical understanding by providing a rigorous analysis of

how structural similarities in transition dynamics enable efficient knowledge transfer. Our results suggest new directions for developing practical algorithms that can systematically leverage shared structure while accounting for task-specific variations.

## 1.1. Related Work

**Transfer RL.** (Agarwal et al., 2023) studied transfer via shared representations between source and target tasks. With generative access to source tasks, they showed that learned representations enable fast convergence to near-optimal policies in target tasks, matching performance as if ground truth features were known. (Cheng et al., 2022) proposed REFUEL for multitask representation learning in low-rank MDPs. They proved that learning shared representations across multiple tasks is more sample-efficient than individual task learning, provided enough tasks are available. Their analysis covers both online and offline downstream learning with shared representations. (Chen et al., 2022; 2024; Chai et al., 2025) analyzed transfer $Q$-learning without transition model assumptions, focusing instead on reward function similarity and transition density. These works established convergence guarantees for both backward and iterative $Q$-learning approaches.

Our work differs by studying transition models with low-rank plus sparse structures. This setting presents *unique challenges* beyond purely low-rank models, as we must identify and leverage an unknown low-rank space while also accounting for sparse deviations.

**Single task RL under structured MDPs.** Single-task RL under structured MDPs has evolved through several key advances: Linear MDPs with known representations were initially studied by (Yang & Wang, 2020), leading to provably efficient online algorithms (Sun et al., 2019; Jin et al., 2020; Zanette et al., 2020; Neu & Pike-Burke, 2020; Cai et al., 2020; Wang et al., 2021).

Low-rank MDPs extend this by requiring representation learning. Major developments include FLAMBE (Agarwal et al., 2020) for explore-then-commit transition estimation, and REP-UCB (Uehara et al., 2022) for balancing representation learning with exploration. Recent work has expanded to nonstationary settings (Cheng et al., 2023) and model-free approaches like MOFFLE (Modi et al., 2024). Related structured models include block MDPs (Du et al., 2019a; Misra et al., 2020; Zhang et al., 2022), low Bellman rank (Jiang et al., 2017), low witness rank (Sun et al., 2019), bilinear classes (Du et al., 2021), and low Bellman eluder dimension (Jin et al., 2021).

Our work introduces the composite MDPs with high-dimensional feature space and low-rank plus sparse transition, extending beyond pure low-rank models. We provide

the first theoretical guarantees for UCB Q-learning under this composite structure.

**Multitask RL and Meta RL.** Research in multitask and meta-RL has evolved through several key theoretical advances. Early work by (Calandriello et al., 2014) examined multitask RL with linear Q-functions sharing sparse support, establishing sample complexity bounds that scale with the sparsity rather than ambient dimension. (Hu et al., 2021) extended this framework by studying weight vectors spanning low-dimensional spaces, showing that sample efficiency improves when the rank is much smaller than both the ambient dimension and number of tasks. (Arora et al., 2020) demonstrated how representation learning reduces sample complexity in imitation learning settings, providing theoretical guarantees for learning shared structure across tasks. (Lu et al., 2022) further developed this direction by analyzing multitask RL with low Bellman error and unknown representations, establishing bounds that improve with task similarity.

Task distribution approaches offered another perspective. (Brunskill & Li, 2013) proved sample complexity benefits when tasks are independently sampled from a finite MDP set, while (Pacchiano et al., 2022) and (Müller & Pacchiano, 2022) extended these results to meta-RL for linear mixture MDPs, showing how learned structure transfers to new tasks. In parallel, research on shared representations by (D'Eramo et al., 2020) established faster convergence rates for value iteration under common structure, and (Lu et al., 2021) proved substantial sample efficiency gains in the low-rank MDP setting. (Zhang et al., 2025) propose a cross-market multi-task dynamic pricing framework that achieves minimax-optimal regret bounds for both linear and nonparametric utilities under structured preference shifts.

Our composite MDP structure advances this line of work by explicitly modeling deviations from low-rank similarity through a sparse component. This framework captures more realistic scenarios where tasks share core structure but maintain individual variations, opening new theoretical directions for multitask and meta-learning approaches.

## 2. Problem Formulation

**Episodic MDPs.** We consider an episodic Markov decision process (MDP) with finite horizon. It is defined by a tuple $\mathcal{M} = (\mathcal{S}, \mathcal{A}, \mathbb{P}, r, \mu, H)$, where $\mathcal{S}$ denotes the state space, $\mathcal{A}$ represents the action space, $H$ is the finite time horizon, $r : \mathcal{S} \times \mathcal{A} \to [0, 1]$ the reward function, $\mathbb{P}$ is the state transition probability, and $\mu$ is the initial state distribution. Let $[H]$ denote the index set $\{1, 2, \cdots, H\}$. A policy $\pi : \mathcal{S} \times [H] \to \mathcal{A}$ maps each state-stage pair to an action that the agent takes in the episode.

For each stage $h \in [H]$, the value function $V_h^\pi : \mathcal{S} \to$

$\mathbb{R}$ evaluates the expected cumulative reward from following policy $\pi$ starting from state $s$ at time $h$, defined as $V_h^\pi(s) = \mathbb{E}\left[\sum_{h'=h}^H r_{h'}(s_{h'}, \pi(s_{h'})) \mid s_h = s\right]$, and $V_{H+1}^\pi(s) = 0$, while the action-value function $Q_h^\pi : \mathcal{S} \times \mathcal{A} \to \mathbb{R}$ evaluates the value of taking action $a$ in state $s$ at time $h$, given by $Q_h^\pi(s, a) = r_h(s, a) + \mathbb{E}\left[\sum_{h'=h+1}^H r_{h'}(s_{h'}, \pi(s_{h'})) \mid s_h = s, a_h = a\right]$ and $Q_H^\pi(s, a) = r_h(s, a)$. The Bellman equation for $V_h^\pi$ and $Q_h^\pi$ can be expressed as $V_h^\pi(s) = Q_h^\pi(s, \pi_h(s))$ and

$$Q_h^\pi(s, a) = r_h(s, a) + [\mathcal{P}V_{h+1}^\pi](s, a),$$

where $[\mathcal{P}V_{h+1}](s, a) := \sum_{s'} \mathbb{P}(s'|s, a)V_{h+1}(s')$[1]. The Bellman optimally equations for the optimal value function and action-value function are as follows:

$$V_h^*(s) = \max_{a \in A}\left\{r_h(s, a) + [\mathcal{P}V_{h+1}^*](s, a)\right\},$$
$$Q_h^*(s, a) = r_h(s, a) + [\mathcal{P}V_{h+1}^*](s, a),$$

with $V_{H+1}^*(s) = 0$ and $Q_H^*(s, a) = r_H(s, a)$.

The **cumulative regret** quantifies the performance discrepancy of an agent over episodes. Given an initial state $s_0 \sim \mu$, for the $n^{th}$ episode, the regret is the value difference of the optimal policy $V^*(s_0)$ and the agent's chosen policy $V^{\pi_n}(s_0)$ which based on its experience up to the beginning of the $n^{th}$ episode and applied throughout the episode. Accumulating over $N$ episodes, it is defined as:

$$\text{Regret}(N) = \sum_{n=1}^N \mathbb{E}_\mu\left[V^*(s_0) - V^{\pi_n}(s_0).\right]$$

The agent aims to learn a sequence of policies $(\pi_1, \ldots, \pi_N)$ to minimize the cumulative regret. If the reward function has a linear feature representation, any additional regret from an unknown reward becomes a lower-order term and does not affect the regret's overall magnitude. For clarity of presentation, we assume the agent knows the reward function and focus primarily on estimating the transition probability.

**Composite MDPs.** Let $\phi(\cdot) \in \mathbb{R}^p$ and $\psi(\cdot) \in \mathbb{R}^q$ be feature functions where $p$ and $q$ can be large. Consider probability transitions $\mathbb{P}(s'|s, a)$ that can be fully embedded in the feature space via a core matrix $M^*$:

$$\mathbb{P}(s'|s, a) = \phi(s, a)^\top \cdot M^* \cdot \psi(s').$$

Since feature dimensions $p$ and $q$ can be large, we need not know the exact feature functions - we can include many possible features to span the space. What matters is learning the structure of $M^*$ from data.

---

[1] Here $\mathcal{P}$ is a function operator mapping from a function $\mathcal{S} \mapsto R$ to a function $\mathcal{S} \times \mathcal{A} \mapsto \mathbb{R}$. One can think of that as a matrix with dimension $SA$ and $S$ in the tabular case.

To capture how transition dynamics combine shared core elements with scenario-specific variations, we impose the following structured assumption on the transition matrix.

**Definition 2.1** (Composite MDPs). A probability transition model $\mathbb{P} : \mathcal{S} \times \mathcal{A} \to \Delta(\mathcal{A})$ can be fully embedded in the feature space characterized by two given feature functions $\phi(\cdot) \in \mathbb{R}^p$ and $\psi(\cdot) \in \mathbb{R}^q$ where both $p$ and $q$ can be large. The core matrix of the transition model decomposes as:

$$\mathbb{P}(s'|s,a) = \phi(s,a)^\top \cdot (L^* + S^*) \cdot \psi(s'),$$

where $L^*$ is a low-rank incoherent matrix and $S^*$ is a sparse matrix.

*Remark* 2.2 (*Distinction from linear and low-rank models*). The composite MDP model generalizes both linear MDPs and low-rank MDPs. Linear MDPs assume a fixed linear transition structure with known feature maps, while low-rank MDPs drop the need for known features but restrict the transition matrix to be entirely low-rank under some feature representation. In contrast, our composite MDP assumes an unknown feature-based factorization for a low-rank core as in the low-rank MDP, but augments it with an additional sparse component. This hybrid structure captures more complex transition dynamics and task-specific deviations than either linear or low-rank models alone, marking a clear departure from those classical assumptions.

*Remark* 2.3 (*Distinction from classical low-rank settings*). Our framework explicitly tackles high-dimensional feature spaces, in contrast to the classical low-rank MDP literature that typically assumes moderate dimensionality (Yang & Wang, 2020; Agarwal et al., 2020). We consider regimes where the feature dimensions ($p$ and $q$) are significantly larger than the number of trajectories ($p, q \gg N$), and crucially, the shared transition core $M^*$ is not constrained to be low-rank. This departure renders existing algorithms for linear or low-rank MDPs inapplicable, since those methods rely on a strictly low-rank structure or low feature complexity. The high-dimensional composite setting therefore demands new estimation techniques that can leverage the mixed low-rank and sparse structure to achieve efficient learning.

*Remark* 2.4 (*Relation to prior models*). The composite MDP shares a structural philosophy with certain prior models – notably the linear mixture MDP frameworks studied by (Yang & Wang, 2019) and (Ayoub et al., 2020) – in that transitions are factorized via feature maps. Our model can be written in the form of (Yang & Wang, 2019) as $\mathbb{P}(s' \mid s, a) = \phi(s,a)^\top \alpha(s')$ with $\alpha(s') := (L^* + S^*)\psi(s')$. This reduces to (Yang & Wang, 2019) when $S^* = 0$. Similarly, our model matches the linear factored MDP in (Ayoub et al., 2020) via a Kronecker formulation: $\mathbb{P}(s' \mid s, a) = (\phi(s,a) \otimes \psi(s'))^\top \text{vec}(L^* + S^*)$. While structurally related, a key difference lies in estimation. Our setting allows $p, q \gg N$, and our estimator achieves minimax-optimal

error rates that do not scale with $d = \max(p, q)$. In contrast, (Yang & Wang, 2019; Ayoub et al., 2020) assume low-dimensional or identifiable parameter spaces and are not suitable for high-dimensional regimes. In addition, our structured assumption is especially crucial in transfer RL, enabling effective knowledge sharing via $L^*$ while capturing task-specific deviations via $S^*$.

## 3. Single-Task UCB-$Q$-Learning under High-Dimensional Composite MDPs

This section introduces UCB-$Q$-Learning for composite MDPs with a high-dimensional feature space. Specifically, we consider the setting where the feature dimensions $p, q \gg N$, and the transition core $M^*$ is no longer a low-rank matrix. As a result, existing algorithms designed for linear MDPs are not applicable. Likewise, methods tailored for low-rank MDPs fail in our setting due to the absence of a low-rank structure in $M^*$. To address the challenges arising from our relaxed dimensionality constraints and the more complex MDP structure, novel algorithmic approaches are required.

For any tuples $(s_{i,h}, a_{i,h})$ from episode $i$ and stage $h$: We define $\phi_{i,h} = \phi(s_{i,h}, a_{i,h})$, $\psi_{i,h} = \psi(s_{i,h})$, and $\boldsymbol{K}_\psi := \sum_{s' \in \mathcal{S}} \psi(s')\psi(s')^\top$. Our estimator is based on the following population-level equation at each stage $h$,

$$
\begin{aligned}
\mathbb{E}\left[\psi_{i,h}^\top \boldsymbol{K}_\psi^{-1} \mid s_{i,h}, a_{i,h}\right] &= \sum_{s'} \mathbb{P}(s'|s_{i,h}, a_{i,h})\psi(s')^\top \boldsymbol{K}_\psi^{-1} \\
&= \sum_{s'} \phi_{i,h}^\top (L^* + S^*)\psi(s')\psi(s')^\top \boldsymbol{K}_\psi^{-1} \\
&= \phi_{i,h}^\top (L^* + S^*). \quad (1)
\end{aligned}
$$

This motivates us to use the sample-level counterpart of (1) to estimate $L^*$ and $S^*$ in a composite MDP. However, both $L^*$ and $S^*$ are unknown. To recover the low-rank and sparse components, additional assumptions are required to ensure that the low-rank part can be separated from the sparse component. Below, we elaborate on the incoherence assumption and sufficient sparsity conditions.

**Assumption 3.1.** Let $L^* = U^*\Sigma^*V^{*T}$ be the singular value decomposition (SVD) of $L^*$. Let $\mu$ be the incoherence parameter (a constant $> 1$) and $r$ be the rank of $L^*$ We assume that:
(i) (Incoherence.) $\|U^*\|_{2,\infty}, \|V^*\|_{2,\infty} \le \sqrt{\frac{\mu r}{p}}$, where the 2-to-infinity norm $\|\cdot\|_{2,\infty}$ denotes the maximum $\ell_2$-norm of the rows of a matrix.
(ii) (Sufficient sparsity.) Matrix $S^*$ contains at most $s$ non-zero entries, where $s \le \overline{s} := \frac{\max\{p,q\}}{4C_S\mu r^3}$, for some constant $C_S$.

*Remark* 3.2 (*Incoherence condition*). The incoherence condition ensures that the singular vectors of a low-rank matrix

are not overly concentrated in any single direction or entry, a property that is crucial for matrix completion (Candes & Recht, 2012). In our setting, it also facilitates the separation of the sparse component from the low-rank matrix. When $r$ and $\mu$ are treated as constants, the maximum permissible sparsity level scales linearly with $p$. Moreover, as shown in (Candès & Tao, 2010), the incoherence condition holds for a broad class of random matrices.

We consider the online learning setting and propose to estimate $L^*$ and $S^*$ in the composite MDP by optimizing the following hard-constrained least-square objective for each episode $n \in [N]$ with collected tuples $(s_{i,h}, a_{i,h})$ from previous episode $i \in [n]$ and stage $h \in [H]$:

$$(\widehat{L}_n, \widehat{S}_n) \in \underset{L,S \in \mathbb{R}^{p \times q}}{\arg\min} \sum_{i \in [n], h \in [H]} \|\psi_{i,h}^\top \boldsymbol{K}_\psi^{-1} - \phi_{i,h}^\top (L+S)\|_2^2$$

$$\text{s.t.} \quad L = U\Sigma V^T, \quad \|U\|_{2,\infty} \le \sqrt{\frac{\mu r}{p}},$$

$$\|V\|_{2,\infty} \le \sqrt{\frac{\mu r}{q}}, \|S\|_0 \le s \tag{2}$$

*Remark* 3.3 (*Computational Efficiency of $\boldsymbol{K}_\psi$*). When $|\mathcal{S}|$ is large or infinite, we can compute $K_\psi$ using a Monte Carlo approximation: $\widehat{K}_\psi = \frac{1}{m} \sum_{i=1}^m \psi(s_i)\psi(s_i)^\top$, $s_i \sim$ Unif($\mathcal{S}$). This is a standard approach in randomized numerical linear algebra (Drineas & Mahoney, 2016) and is computed only once before online learning. Our method is thus comparable in efficiency to (Yang & Wang, 2019), which stores empirical covariances, but we use $K_\psi$ to build confidence regions specific to our model.

### 3.1. UCB-$Q$ Learning for High-Dimensional Composite MDPs

Since the transition dynamics $\mathbb{P}$ are typically unknown, we must leverage observed data to approximate the underlying model parameters. To balance the exploration-exploitation trade-off, we adopt the optimism-in-the-face-of-uncertainty principle by employing an Upper Confidence Bound (UCB)-based algorithm. We begin by constructing the confidence region:

$$\mathcal{B}_n = \{(L,S) \mid \left\|L - \widehat{L}_n\right\|_F^2 + \left\|S - \widehat{S}_n\right\|_F^2 \le \beta_n\} \tag{3}$$

where $\widehat{L}_n$ and $\widehat{S}_n$ are estimated by (2),

$$\beta_n = \frac{C_\beta H \log(dNH)}{n} \left(r(C_\phi C_\psi')^2 + s C_{\phi\psi}^2\right), \tag{4}$$

$d = \max\{p, q\}$ and $C_\phi, C_\psi, C_\psi', C_{\phi\psi}$ are positive parameters defined in the regularity Assumption 3.4, $C_\beta$ is a universal constant. The optimistic value functions are given

---

**Algorithm 1** UCB-$Q$ Learning for HD Composite MDPs

**Input:** Total number of episodes $N$, feature function $\phi \in \mathbb{R}^p, \psi \in \mathbb{R}^q$.
**for** episode $n = 1, 2, \ldots, N$ **do**
  Construct confidence region in (3)
  Calculate $Q_{n,h}(s, a)$ in (5)
  **for** stage $h = 1, 2, \ldots, H$ **do**
    Take action $a_{n,h} = \arg\max_{a \in \mathcal{A}} Q_{n,h}(s_{n,h}, a)$
    Observe next state $s_{n,h+1}$
  **end for**
  Learn transition core estimator $\widehat{L}_n, \widehat{S}_n$ using (2).
**end for**

---

by:

$$Q_{n,h}(s,a) = r(s,a) + \max_{L,S \in \mathcal{B}_n} \phi(s,a)^\top (L+S)\Psi^\top V_{n,h+1},$$
$$\tag{5}$$
$$Q_{n,H+1}(s,a) = 0,$$

where $V_{n,h}(s) = \Pi_{[0,H]}[\max_a Q_{n,h}(s,a)]$, with $\Pi_{[0,H]}$ truncating values to $[0, H]$. Here, $\Psi \in |\mathcal{S}| \times q$ is feature matrix, where each row represents the $q$-dimensional feature vector corresponding to a unique state in the state space $\mathcal{S}$. The algorithm is summarized in Algorithm 2.

### 3.2. Regret Analysis for UCB-Q-Learning under High-Dimensional Composite MDPs

For the regret analysis, we impose certain regularity conditions on the features as outlined below.

**Assumption 3.4.** Let $\Psi$ be a matrix with rows as $\psi(s)^\top$. Let $C_\phi, C_\psi, C_\psi', C_{\phi\psi}$ be positive parameters such that

(i) $\forall (s,a), \quad \|\phi(s,a)\|_2 \le C_\phi, \ \|\phi(s,a)\|_\infty \le C_\phi'$;

(ii) $\|\Psi\|_{2,\infty} \le C_\psi$;

(iii) $\forall s', \quad \|\psi(s')^\top \boldsymbol{K}_\psi^{-1}\|_2 \le C_\psi'$;

(iv) $\forall (s,a,s'), \quad \|\phi(s,a)^\top \psi(s') \boldsymbol{K}_\psi^{-1}\|_{\max} \le C_{\phi\psi}$.

**Lemma 3.5** (Transition Estimation Error)**.** *For composite MDPs in Definition 2.1, under Assumption 3.1 and 3.4, the estimator obtained by solving program (2) at the end of $n^{th}$-episode satisfies, with probability at least $1 - 1/(n^2 H)$, that,*

$$\left\|\widehat{L}_N - L^*\right\|_F^2 + \left\|\widehat{S}_N - S^*\right\|_F^2 \le \beta_n,$$

*where $\beta_n$ is defined in (4).*

*Remark* 3.6. Our method handles the high-dimensional setting by explicitly exploiting the low-rank-plus-sparse structure of the transition core matrix $M^* = L^* + S^*$, where $L^*$ is of low-rank $r << d$ and $S^*$ is of sparsity $s << d$. This structure enables consistent estimation in regimes where

$p, q \gg N$, and is key to our theoretical guarantees. This error bound is minimax optimal with respect to $n$.

**Theorem 3.7** (Single-Task Regret Upper Bound). *For composite MDPs in Definition 2.1, under Assumption 3.1 and 3.4, let* Regret(NH) *be the accumulative regret of a total of $N$ episodes using the UCB-Q-Learning in Algorithm 2. We have that*

$$\text{Regret(NH)} \leq C_\phi C_\psi H^2 \sum_{n=1}^{N} \sqrt{2\beta_n} + 1$$
$$\lesssim C_{reg} \sqrt{NH^5}$$

*where $d = \max\{p, q\}$ and*

$$C_{reg} := C_\phi C_\psi \sqrt{C_\beta \left( r(C_\phi C'_\psi)^2 + s C_{\phi\psi}^2 \right) \log(dNH)}.$$

*Remark* 3.8. This regret bound achieves optimal scaling with respect to both the number of trajectories $N$ and ambient dimension $d$, matching previous results in reinforcement learning (Yang & Wang, 2020; Jin et al., 2020). In Section 4, we demonstrate that transfer learning can substantially reduce both the dependence on ambient dimension $d$ and the scaling with $N$ by effectively utilizing additional trajectories from a source task.

### 3.3. Challenge and Proof Sketch under the Composite Structure

By optimality condition of (2), it holds that

$$\sum_{i<n,h\leq H} \|\psi_{i,h}^\top \boldsymbol{K}_\psi^{-1} - \phi_{i,h}^\top(\widehat{L} + \widehat{S})\|_2^2$$
$$\leq \sum_{i<n,h\leq H} \|\psi_{i,h}^\top \boldsymbol{K}_\psi^{-1} - \phi_{i,h}^\top(L^* + S^*)\|_2^2$$

Expanding the inequality, we have

$$\sum_{i<n,h\leq H} \|\phi_{i,h}^\top(\widehat{L} - L^*)\|^2 + \|\phi_{i,h}^\top(\widehat{S} - S^*)\|_2^2 \leq$$
$$2 \sum_{i<n,h\leq H} \langle \phi_{i,h}^\top(L^* + S^* - \widehat{L} - \widehat{S}), \psi_{i,h}^\top \boldsymbol{K}_\psi^{-1} - \phi_{i,h}^\top M^* \rangle$$
$$- 2 \sum_{i<n,h\leq H} \langle \phi_{i,h}^\top(\widehat{L} - L^*), \phi_{i,h}^\top(\widehat{S} - S^*) \rangle$$

Establishing Theorem 3.7 presents several challenges and requires new techniques. First, deriving a high-probability error bound for $\widehat{L}$ and $\widehat{S}$ is nontrivial due to the presence of cross terms at the end of the inequality. To address this, we adapt the separation lemma from (Chai & Fan, 2024), which provides a way to control these cross terms effectively.

Second, ensuring the strong convexity of the linear operator is challenging due to high correlations across stages. To overcome this, we enforce the strong convexity property by incorporating a restart mechanism for each trajectory.

Thirdly, we must bound the error term $\sum_{i=1}^{n-1} \sum_{h=1}^{H} \phi_{i,h} \left( \psi_{i,h}^\top \boldsymbol{K}_\psi^{-1} - \phi_{i,h}^\top M^* \right)$. Since this term forms a martingale difference sequence, we apply matrix concentration techniques to control it effectively.

## 4. Transition Transfer under Composite MDPs

In this section, we consider transfer learning with target task $\mathcal{M}^{*(1)}$ and source task $\mathcal{M}^{*(0)}$. The transition probabilities of the target and source tasks are, respectively,

$$\mathbb{P}^{(0)}(s'|s, a) = \phi(s, a)^\top M^{*(0)} \psi(s'), \quad \text{and}$$
$$\mathbb{P}^{(1)}(s'|s, a) = \phi(s, a)^\top M^{*(1)} \psi(s'), \tag{6}$$

where the core transition matrices $M^{(1)}$ and $M^{(0)}$ are different.

We propose modeling task similarity through their transition dynamics: similar tasks share a common low-rank structure capturing core dynamics, while differing only in sparse directions that represent task-specific variations.

**Assumption 4.1** (Transition Similarity). Consider the target and source tasks characterized by transition model (6). The target and source tasks are different in that their core transition matrices $M^{*(1)} \neq M^{*(0)}$. However, their similarity is defined by:

$$M^{*(0)} = L^* + S^{*(0)}, \quad \text{and} \quad M^{*(1)} = L^* + S^{*(1)}, \tag{7}$$

where both tasks share the same low-rank component $L^*$, $\|S^{*(0)}\|_0 = s_0$, $\|S^{*(1)}\|_0 = s_1$ are task-specific sparse components, two tasks are similar in the sense that their difference $D^* = S^{(1)} - S^{(0)}$, called the "sparsity difference", is very sparse: $\|D^*\|_0 = e \ll \max\{s_0, s_1\}$.

We have $N_0$ episodes for the source task and $N_1 = N$ episodes for the target task. In practice, $N_0 \gg N$ and we would like to use the source task to enhance the performance of the target task. Since our primary focus is on the target data, we don't make specific data generating assumptions on the source data which can be both batch data or generated from certain online process.

For notation brevity, we use $i$ and $h$ to index episodes and time steps of the source task, with $i \in [N_0]$ and $h \in [H]$. For the target task, we use $j$ and $h$ to index its episodes and time steps, with $j \in [N]$ and $h \in [H]$. We denote the following state-action-station transition triplet: $(s_{i,h}, a_{i,h}, s'_{i,h})$ from the target task and $(s_{j,h}, a_{j,h}, s'_{j,h})$ from the source task. The associated features are

$$\phi_{i,h}^{(0)} := \phi(s_{i,h}, a_{i,h}) \in \mathbb{R}^p, \quad \psi_{i,h}^{(0)} := \psi(s'_{i,h}) \in \mathbb{R}^q,$$
$$\phi_{j,h}^{(1)} := \phi(s_{j,h}, a_{j,h}) \in \mathbb{R}^p, \quad \psi_{j,h}^{(1)} := \psi(s'_{j,h}) \in \mathbb{R}^q. \tag{8}$$

Let $\boldsymbol{K}_\psi := \sum_{s' \in \mathcal{S}} \psi(s')\psi(s')^\top$. We have, at each step $h$ for the target task,

$$\mathbb{E}\left[\phi_{i,h}^{(1)}\psi_{i,h}^{(1)\top}\boldsymbol{K}_\psi^{-1} \mid s_{i,h}, a_{i,h}\right] = \left(\phi_{i,h}^{(1)}\phi_{i,h}^{(1)\top}\right)(L^* + S^{*(0)}).$$

Similarly, we have for the source task,

$$\mathbb{E}\left[\phi_{j,h}^{(0)}\psi_{j,h}^{(0)\top}\boldsymbol{K}_\psi^{-1} \mid s_{j,h}, a_{j,h}\right] = \left(\phi_{j,h}^{(0)}\phi_{j,h}^{(0)\top}\right)(L^* + S^{*(1)}).$$

### 4.1. UCB Transfer $Q$-Learning for High-Dimensional Composite MDPs

Now we introduce the UCB Transfer $Q$-Learning (UCB-TQL) for HD Composite MDPs. The algorithm is summarized in Algorithm 2. We first introduce the optimization-based estimator in the following two steps, then proceed to construct the confidence region.

STEP I. Estimate the low-rank and sparse components of the source task by solving[2]

$$(\widehat{L}, \widehat{S}^{(0)}) \in \underset{L,S \in \mathbb{R}^{p \times q}}{\arg\min} \sum_{i \leq N_0, h \leq H} \left\| \psi_{i,h}^{(0)\top}\boldsymbol{K}_\psi^{-1} - \phi_{i,h}^{(0)\top}(L+S) \right\|_2^2$$

$$\text{s.t.} \quad L = U\Sigma V^T, \quad \|U\|_{2,\infty} \leq \sqrt{\frac{\mu r}{p}},$$

$$\|V\|_{2,\infty} \leq \sqrt{\frac{\mu r}{q}}, \|S\|_0 \leq s_0. \tag{9}$$

STEP II. Use target data to correct the bias of the sparse part in an online fashion.

$$\widehat{D}_n \in \underset{D \in \mathbb{R}^{p \times q}}{\arg\min} \sum_{j < n, h \leq H} \left\| \psi_{j,h}^{(1)\top}\boldsymbol{K}_\psi^{-1} - \phi_{j,h}^{(1)\top}(\widehat{L}+\widehat{S}^{(0)}+D) \right\|_2^2$$

$$\text{s.t.} \quad \|D\|_0 \leq e \tag{10}$$

Then the target estimator for $n$ episode is given by

$$(\widehat{L}_n, \widehat{S}_n^{(1)}) = (\widehat{L}, \widehat{S}^{(0)} + \widehat{D}_n). \tag{11}$$

To construct the confidence region, suppose at the first stage, we established $\|L^* - \widehat{L}\|_F^2 + \|S^{*(0)} - \widehat{S}^{(0)}\|_F^2 \leq \beta_{N_0}$ with probability at least $1 - 1/(2N^2H)$, where we slightly abuse notation by again referring to $\beta_{N_0}$ as the confidence radius at initial stage of target learning. When the source samples come from the online UCB algorithm as described in Section 3, we have

$$\beta_{N_0} = \frac{C_\beta H \log(dN_0 H)}{N_0}\left(r(C_\phi C_\psi')^2 + sC_{\phi\psi}^2\right), \tag{12}$$

$d = \max\{p, q\}$ and $C_\phi, C_\psi, C_\psi', C_{\phi\psi}$ are positive parameters defined in the regularity Assumption 3.4, $C_\beta$ is a universal constant.

[2]Note that for simplicity, we assume the sparsity $s_0$ appearing in the constraint is known. It can be replaced by an upper bound on $s_0$.

---

**Algorithm 2** UCB-TQL for Composite MDPs

**Input:** $N_0$ episodes of source data, feature function $\phi \in \mathbb{R}^p, \psi \in \mathbb{R}^q$, number of episodes $N$ of the target task. Calculate pilot transition core estimators $\widehat{L}, \widehat{S}^{(0)}$ using (9).

**for** episode $n = 1, 2, \ldots, N$ **do**
  Construct confidence region (13).
  Calculate $Q_{n,h}(s, a)$ in (14).
  **for** stage $h = 1, 2, \ldots, H$ **do**
    Take action $a_{n,h} = \arg\max_{a \in \mathcal{A}} Q_{n,h}(s_{n,h}, a)$
    Observe $s_{n,h+1}$ from target domain $\mathcal{M}^{(1)}$
  **end for**
  Learn transition core estimator using (11).
**end for**

---

The online confidence region at step $n$ is then constructed as

$$\widetilde{\mathcal{B}}_n = \left\{ \begin{array}{l} (L, S, D) : \|L - \widehat{L}^{(0)}\|_F^2 + \|S - D - \widehat{S}^{(0)}\|_F^2 \leq \beta_{N_0}, \\ \|D - \widehat{D}_n\|_F^2 \leq \beta_n^{(1)}, \quad \|D\|_0 \leq e \end{array} \right\}. \tag{13}$$

where we incorporate $D$ in the decision variables to put direct restriction on the sparsity of sparse difference.

Similarly, the optimistic value functions are calculated as follows.

$$Q_{n,h}(s, a) = r(s, a) + \max_{L,S \in \widetilde{\mathcal{B}}_n} \phi(s, a)^\top(L+S)\Psi^\top V_{n,h+1}, \tag{14}$$

$$Q_{n,H+1}(s, a) = 0,$$

*Remark* 4.2 (*Extensions*). We focus on sparsity-constrained optimization, which can be extended to a Lasso-type $L_1$ penalty for improved computational efficiency. For brevity, we omit these details here.

### 4.2. Regret Analysis of UCB-TQL

The following assumption is in parallel to Assumption 3.1.

**Assumption 4.3.** Consider transfer RL setting with transition similarity defined in Assumption 4.1. Recall that $L^* = U^*\Sigma^*V^*$. We assume that $\|U^*\|_{2,\infty}, \|V^*\|_{2,\infty} \leq \sqrt{\frac{\mu r}{p}}$ and that the sparsity of $S^{*(0)}$ and $S^{*(1)}$ satisfies $\max\{s_0, s_1\} \leq \bar{s} := \frac{\max\{p, q\}}{4C_S \mu r^3}$, for some constant $C_S$.

The following theorem demonstrates the provable benefits of UCB-TQL for the target RL task.

**Lemma 4.4** (Estimation Error). *Let $N_0$ denotes the number of episodes from the source task. Under Assumption 3.4, 4.1, and 4.3, the estimator at the end of $n^{th}$-episode satisfies with probability at least $1 - 1/(n^2 H)$ that,*

$$\left\|\widehat{L}_n - L^*\right\|_F^2 + \left\|\widehat{S}_n - S^{*(0)}\right\|_F^2 \leq \beta_{N_0} + \frac{eC_{\phi\psi}^2 H \log(dnH)}{n},$$

*where $\beta_{N_0}$ is the initial confidence radius defined in* (12).

*Remark* 4.5. The estimation error bound is minimax optimal with respect to $N_0$, $n$, and $d$. We extend these existing results in the contexts of regression and matrix completion (Chai & Fan, 2024) to the settings of reinforcement learning and transfer learning.

**Theorem 4.6** (Regret upper bound for UCB-TQL). *Let $N_0$ and $N$ denotes the number of episodes from the source and target tasks, respectively. Let* Regret(NH) *be the accumulative regret of a total of $N$ target episodes using the UCB-TQL in Algorithm 2. Under Assumption 3.4, 4.1, and 4.3, it holds that*

$$\text{Regret(NH)} \lesssim C'_{reg} N / \sqrt{N_0} + C'_\phi C_\psi H^2 \sqrt{eC^2_{\phi\psi} NH \log{(dNH)}}$$

*where $s = \max\{s_0, s_1\}$ and*

$$C'_{reg} := \left(C_\phi + C'_\phi \sqrt{e}\right) C_\psi \sqrt{C_\beta H^5 \log(dN_0 H) \left(r(C_\phi C'_\psi)^2 + sC^2_{\phi\psi}\right)}. \tag{15}$$

*Remark* 4.7. Note that the first term represents the rate at which the source is learned, while the second term accounts for correcting the bias of the sparse component.

When the source sample size $N_0$ is sufficiently larger than the target sample size, the regret is dominated by the second term. Specifically, when $N_0 \asymp N^2$, the regret bound simplifies to $\widetilde{\mathcal{O}}(\sqrt{eH^5 N})$, which scales independently of the ambient dimension. Since $e \ll d$, this represents a significant improvement over the result in (Yang & Wang, 2020).

We also characterize the phase transition. Specifically, when $N_0 \geq N(rC^2_\phi + s)$, neglecting the logarithm terms, the regret bound becomes dominated by the second term, corresponding to estimation of the sparse difference.

### 4.3. Challenges and Proof Sketch of UCB-TQL with High-Dimensional Composite MDPs.

A natural way to construct the confidence region is

$$\mathcal{B}_n = \left\{ (L, S) \mid \left\|L - \widehat{L}_n\right\|^2_F + \left\|S - \widehat{S}_n\right\|^2_F \leq \beta^{(1)}_n \right\} \tag{16}$$

where $\beta^{(1)}_n := \beta_{N_0} + \frac{eC^2_{\phi\psi} H \log(dNH)}{n}$.

However, this confidence region is not tight in that we are not fully utilizing the sparse difference $D$. To be more

specific, plugging the value of $\beta^{(1)}_n$ in (19), we have

$$\text{Regret(NH)} \lesssim C_\phi C_\psi H \sum_{n=1}^{N} \sqrt{\beta^{(1)}_n} + 1$$
$$\lesssim C_\phi C_\psi H \Big( N\sqrt{\beta_{N_0}} + \sqrt{e\, C^2_{\phi\psi} \, NH \, \log(dNH)} \Big).$$

In contrast to (16), we employ a more fine-grained confidence region (13), where we directly restrict the sparsity of the sparse difference $D$ to be bounded, leading to improved rates.

In particular, we have $(L^*, S^{*(0)}, D^*) \in \widetilde{\mathcal{B}}_n$, indicating this CR is valid. To bound the one-step error, let $(\widetilde{L}, \widetilde{S}, \widetilde{D}) = \arg\max_{L,S \in \widetilde{\mathcal{B}}_n} \phi(s,a)^\top (L + S)\Psi^\top V_{n,h+1}$, it holds that

$$Q_{n,h}(s_{n,h}, a_{n,h}) - (r(s_{n,h}, a_{n,h}) + [P_h V_{n,h+1}](s_{n,h}, a_{n,h}))$$
$$\leq \left\|\phi^\top_{n,h}(\widetilde{L} - L^*)\right\|_2 \left\|\Psi^\top V_{n,h+1}\right\|_2 +$$
$$\left\|\phi^\top_{n,h}\left(\widetilde{S} - \widetilde{D} - S^* + D^*\right)\right\|_2 \left\|\Psi^\top V_{n,h+1}\right\|_2 +$$
$$\left|\phi^\top_{n,h}\left(\widetilde{D} - D^*\right)\Psi^\top V_{n,h+1}\right| \tag{17}$$

The first two terms can be bounded similar to single-task case. From the constraint in the optimization problem (9) and Assumption 4.3, we have $\|\widetilde{D}_n\|_0 \leq e$, $\|D^*\|_0 \leq e$, implying $\|\widetilde{D}_n - D^*\|_0 \leq 2e$. This observation facilitates a tight bound on the third term. Combining these one-step error bounds then yields the final regret bound in (19).

## 5. Discussion

When employing low-rank and sparse structures as the core for transition probabilities, several directions for future exploration emerge. Firstly, alternative sparse structures, such as row sparsity, column sparsity, or group sparsity, could be further investigated to understand their impact on learning dynamics and efficiency. These alternative formulations may offer more nuanced or efficient ways to capture the underlying patterns in transition dynamics across different domains.

Secondly, our analysis reveals that the regret bounds of the UBC-TQL algorithm are significantly influenced by the error bounds derived from matrix recovery. Since the Upper Confidence Bound (UCB) is determined by the error bounds of matrix recovery, the regret bound is largely dictated by these errors. An extension goal is to achieve the current levels of regret under more relaxed assumptions. This could involve developing new theoretical frameworks or algorithms that either provide tighter error bounds or leverage additional structure in the transition dynamics that has not been fully exploited.

## Acknowledgements

Elynn Chen's research is supported in part by the NSF Award 2412577. Lin Yang's research is supported in part by the NSF Award 2221871 and an Amazon Faculty Award.

## Impact Statement

This work addresses a fundamental gap between the empirical success of transfer reinforcement learning and its limited theoretical foundation, while presenting a general approach applicable across robotics, autonomous systems, and other complex decision-making environments. Our core contribution is a novel composite MDP framework in which each task's transition dynamics are modeled as the sum of a low-rank shared structure and a sparse task-specific component. Leveraging this formulation, we develop UCB-TQL (Upper Confidence Bound Transfer Q-Learning), a new algorithm with provable regret bounds and rigorous structural performance guarantees. By uniting theoretical guarantees with broad real-world applicability, this research bridges the gap between practical transfer learning and formal analysis, laying the groundwork for more efficient and principled reinforcement learning across diverse domains.

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

# Supplemental Materials

**Notation** We use $[h] = \{1, 2, \ldots, h\}$ for integers from 1 to $h$. In this paper, vectors are assumed to be column vectors. For a vector $\mathbf{v} \in \mathbb{R}^p$, the norms $\|\mathbf{v}\|_1$, $\|\mathbf{v}\|_2$, and $\|\mathbf{v}\|_\infty$ represent the 1-norm, Euclidean norm(or 2-norm), and infinity norm, respectively. For a matrix $M \in \mathbb{R}^{p \times q}$, we use the following notation for norms: $\|M\|_0$ denotes the number of non-zero elements, $\|M\|_1 = \sum_{i=1}^p \sum_{j=1}^q |M_{ij}|$ is the sum of the absolute values of all elements, and $\|M\|_{\max} = \max_{i,j} |M_{ij}|$ is the maximum absolute value among elements. The Frobenius norm is $\|M\|_F = \sqrt{\mathrm{Tr}(M^\top M)} = \sqrt{\sum_{j=1}^{d_1} \sum_{k=1}^{d_2} M_{jk}^2}$, which is also equivalent to $\sqrt{\sum_j \sigma_j(M)^2}$, where $\sigma_j(M)$ are the singular values of $M$. The nuclear norm is $\|M\|_* = \sum_j \sigma_j(M)$, and the operator norm is $\|M\|_{\mathrm{op}} = \max_j \sigma_j(M)$. Moreover, the $2-\mathrm{to}-\infty$ norm is defined as $\|M\|_{2,\infty} = \max_{j=1}^{d_1} \|M_{j,:}\|_2$ For two matrices, $\langle L, S \rangle$ represents the Euclidean inner product. We use $a_n = \mathcal{O}(b_n)$ or $a_n \lesssim b_n$ if there exists some $C > 0$ such that $a_n \leq C b_n$. $\widetilde{\mathcal{O}}(\cdot)$ is similarly defined, neglecting logarithmic factors. Constants $c, C, c_0, \cdots$ may vary from line to line.

# A. Regret Analysis of the Single-Task UCB-$Q$-Learning with Composite MDP Structures

We present below the proof of Theorem 3.7. The proof of Lemma 3.5 emerges as an intermediate step along the way.

**Proof of Theorem 3.7.** We first sketch the proof as follows. First of all, we define the "good event" that the ground truth transition core matrix before episode $n$ lies in the confidence region as $\mathcal{E}_n$, i.e., $(L^*, S^*) \in \mathcal{B}_{n'}$ for any $n' \leq n - 1$. We assume $\mathcal{E}_n$ holds first and use concentration to prove that $\mathcal{E}_n$ holds with high probability later. We denote $E_n = \mathbb{1}_{\mathcal{E}_n}$.

1. Under $\mathcal{E}_n$, prove $Q_{n,h} \geq Q_h^*$ using induction.

2. Bound $Q_{n,h}(s_{n,h}, a_{n,h}) - [r(s_{n,h}, a_{n,h}) + P(\cdot|s_{n,h}, a_{n,h})^T V_{n,h+1}]$.

3. Bound the total regret by one-step errors derived in Step 2.

We elaborate each step in the sequel.

## A.1. Upper confidence bound

**Lemma A.1.** *Given any state-action pair $(s, a) \in \mathcal{S} \times \mathcal{A}$, for each episode $n$ and decision step $h$, we have:*

$$Q_{n,h}(s, a) \geq Q_h^*(s, a).$$

*Proof.* We use induction. At $h = H$, we have $Q_{n,H} = Q_H^*(s, a) = r(s, a)$. Assuming the argument is true for $1 < h' \leq H$, it naturally extends to $h = h' - 1$ that $V_{n,h}(s) \geq V_h^*(s)$, and hence

$$Q_{n,h}(s, a) = r(s, a) + \max_{L, S \in \mathcal{B}_n} \phi(s, a)^\top (L + S) \Psi^\top V_{n,h+1}$$

$$\geq r(s, a) + \phi(s, a)^\top (L^* + S^*) \Psi^\top V_{n,h+1}$$

$$\geq r(s, a) + [PV_{h+1}^*](s, a) = Q_h^*(s, a).$$

$\square$

## A.2. One-step bound

**Lemma A.2.** *For any $h \in [H]$ and $n \in [N]$, we have*

$$Q_{n,h}(s_{n,h}, a_{n,h}) - (r(s_{n,h}, a_{n,h}) + [P_h V_{n,h+1}](s_{n,h}, a_{n,h})) \leq C_\phi C_\psi H \sqrt{2\beta_n}.$$

*Proof.* Let $(\widetilde{L}, \widetilde{S}) = \arg\max_{L, S \in \mathcal{B}_n} \phi(s, a)^\top (L + S) \Psi^\top V_{n,h+1}$, it holds that

$$Q_{n,h}(s_{n,h}, a_{n,h}) - (r(s_{n,h}, a_{n,h}) + [P_h V_{n,h+1}](s_{n,h}, a_{n,h})) = \phi_{n,h}^\top \left[ \left( \widetilde{L} + \widetilde{S} \right) - M^* \right] \Psi^\top V_{n,h+1}$$

$$= \phi_{n,h}^\top \left[ (\widetilde{L} - L^*) + \left( \widetilde{S} - S^* \right) \right] \Psi^\top V_{n,h+1}.$$

Applying Hölder's inequality, the triangle inequality, and Cauchy-Schwarz inequality, we deduce the following results:

$$
\begin{aligned}
Q_{n,h}&(s_{n,h}, a_{n,h}) - (r(s_{n,h}, a_{n,h}) + [P_h V_{n,h+1}](s_{n,h}, a_{n,h})) \\
&\leq \left\| \phi_{n,h}^\top (\widetilde{L} - L^*) \right\|_2 \left\| \Psi^\top V_{n,h+1} \right\|_2 + \left\| \phi_{n,h}^\top \left( \widetilde{S} - S^* \right) \right\|_2 \left\| \Psi^\top V_{n,h+1} \right\|_2 \\
&\leq H \left( C_\psi \left\| \phi_{n,h}^\top (\widetilde{L} - L^*) \right\|_2 + C_\psi \left\| \phi_{n,h}^\top \left( \widetilde{S} - S^* \right) \right\|_2 \right) \\
&\leq H \left( C_\psi \left\| \phi_{n,h} \right\|_2 \left\| \widetilde{L} - L^* \right\|_F + C_\psi \left\| \phi_{n,h} \right\|_2 \left\| \widetilde{S} - S^* \right\|_F \right) \\
&\leq C_\phi C_\psi H \left( \left\| \widetilde{L} - L^* \right\|_F + \left\| \widetilde{S} - S^* \right\|_F \right) \\
&\leq C_\phi C_\psi H \sqrt{2\beta_n}.
\end{aligned}
\tag{18}
$$

$\square$

## A.3. Regret decomposition

We bound the regret by the sum of one-step errors. To set the stage, let $\mathcal{F}_{n,h}$ be defined as the $\sigma$-field generated by all the random variables up until episode $n$, step $h$, essentially fixing the sequence $s_{1,1}, a_{1,1}, s_{1,2}, a_{1,2}, \ldots, s_{n,h}, a_{n,h}$. To proceed, let

$$
\delta_{n,h} := (V_{n,h} - V_h^{\pi_n})(s_{n,h}), \quad \gamma_{n,h} := Q_{n,h}(s_{n,h}, a_{n,h}) - (r(s_{n,h}, a_{n,h}) + [P_h V_{n,h+1}](s_{n,h}, a_{n,h})).
$$

And hence

$$
\begin{aligned}
\mathrm{Regret}(NH) &= \mathbb{E} \left( \sum_{n=1}^N [V^*(s_0) - V^{\pi_n}(s_0)] \right) \\
&\leq \mathbb{E} \left( \sum_{n=1}^N (V_{n,1} - V_1^{\pi_n})(s_0) \right) = \sum_{n=1}^N \mathbb{E}(\delta_{n,1}).
\end{aligned}
$$

We have

$$
\mathbb{E}(\delta_{n,1}) = \mathbb{E}(\delta_{n,1} E_n + (1 - E_n)\delta_{n,1}) \leq \mathbb{E}(\delta_{n,1} E_n) + H\mathbb{P}[E_n = 0]
$$

and

$$
\begin{aligned}
\mathbb{E}\left( \delta_{n,1} E_n \mid \mathcal{F}_{n,1} \right) &= \left( V_{n,1} - V_1^{\pi_n} \right)(s_{n,1}) E_n \\
&\leq Q_{n,1}(s_{n,1}, a_{n,1}) - V_1^{\pi_n}(s_{n,1}) \\
&= \gamma_{n,1} + \left( r(s_{n,1}, a_{n,1}) + [PV_{n,2}](s_{n,1}, a_{n,1}) \right) - V_1^{\pi_n}(s_{n,1}) \\
&\leq \gamma_{n,1} + \mathbb{E}\left( (V_{n,2} - V_2^{\pi_n})(s_{n,2}) E_n \mid \mathcal{F}_{n,1} \right) \leq \cdots \\
&\leq \sum_{h=1}^H \mathbb{E}(\gamma_{n,h} \mid \mathcal{F}_{n,1}) \\
&\leq C_\phi C_\psi H^2 \sqrt{2\beta_n}.
\end{aligned}
$$

Therefore, we establish the regret bound with high probability that

$$
\mathrm{Regret}(NH) \leq C_\phi C_\psi H^2 \sum_{n=1}^N \sqrt{2\beta_n} + NH\mathbb{P}[E_N = 0],
\tag{19}
$$

where we use $\mathcal{E}_n \subseteq \mathcal{E}_{n'}$ with $n > n'$.

## A.4. Confidence region

In this subsection, we validate the confidence region. Recall the CR is defined in (3).

Denote $X_n = \begin{bmatrix} \phi_{1,1}^\top \\ \cdots \\ \phi_{1,H}^\top \\ \cdots \\ \phi_{n-1,H}^\top \end{bmatrix}$ and $Y_n = \begin{bmatrix} \psi_{1,1}^\top \boldsymbol{K}_\psi^{-1} \\ \cdots \\ \psi_{1,H}^\top \boldsymbol{K}_\psi^{-1} \\ \cdots \\ \psi_{n-1,H}^\top \boldsymbol{K}_\psi^{-1} \end{bmatrix}$, we have the observational model as follows:

$$Y_n = X_n(L^* + S^*) + W_n$$

where $W_n = Y_n - X_n(L^* + S^*)$.

In view of (Chai & Fan, 2024), we need the observation model to satisfy the restricted strong convexity condition in order for the low-rank part and sparse part to be separated.

$$\lambda_{\min}\left(\frac{X_n^\top X_n}{n-1}\right) \geq c_1. \tag{20}$$

We will later give specific cases in which this inequality holds. In the sequel, we bound $\|X_n^\top W_n\|_2$ and $\|X_n^\top W_n\|_{\max}$.

In fact, we can express $X_n^\top W_n$ as

$$X_n^\top W_n = \sum_{i=1}^{n-1} \sum_{h=1}^{H} \phi_{i,h}\left(\psi_{i,h}^\top \boldsymbol{K}_\psi^{-1} - \phi_{i,h}^\top M^*\right).$$

Note that $\mathbb{E}\left[\psi_{i,h}^\top \boldsymbol{K}_\psi^{-1} - \phi_{i,h}^\top M^* | \mathcal{F}_{i,h}\right] = 0$, and $X_n^\top W_n$ is a sum of martingale differences. Let $Z_{i,h} = \phi_{i,h}\left(\psi_{i,h}^\top \boldsymbol{K}_\psi^{-1} - \phi_{i,h}^\top M^*\right)$, we have that

$$\begin{aligned}
\|Z_{i,h}\|_2 &= \|\phi_{i,h}\| \cdot \|\psi_{i,h}^\top \boldsymbol{K}_\psi^{-1} - \phi_{i,h}^\top M^*\| \\
&= \|\phi_{i,h}\| \cdot \left(\|\psi_{i,h}^\top \boldsymbol{K}_\psi^{-1}\| + \|\phi_{i,h}^\top M^*\|\right) \\
&\leq 2C_\phi C_\psi',
\end{aligned}$$

where we used $\|\phi_{i,h}^\top M^*\| = \left\|\mathbb{E}\left[\psi_{i,h}^\top \boldsymbol{K}_\psi^{-1} | \mathcal{F}_{i,h}\right]\right\| \leq C_\psi'$.

On the other hand,

$$\left\|\sum_{i=1}^{n-1}\sum_{h=1}^{H} \mathbb{E}[Z_{i,h}^\top Z_{i,h} | \mathcal{F}_{i,h}]\right\| \leq 4nH(C_\phi C_\psi')^2$$

and

$$\left\|\sum_{i=1}^{n-1}\sum_{h=1}^{H} \mathbb{E}[Z_{i,h} Z_{i,h}^\top | \mathcal{F}_{i,h}]\right\| \leq 4nH(C_\phi C_\psi')^2.$$

By Matrix Freedman inequality (Corollary 1.3 in (Tropp, 2011)), we have with probability at least $1 - \delta$ that

$$\|X_n^\top W_n\|_2 \lesssim C_\phi C_\psi' \log\left(\frac{d}{\delta}\right) + C_\phi C_\psi' \sqrt{nH \log\left(\frac{d}{\delta}\right)}.$$

Similarly, each entry of $X_n^\top W_n$ is the sum of martingale differences, almost surely bounded by $2C_{\phi\psi}$, and we have by Azuma-Hoeffding's inequality and a union bound over $d^2$ entries that, with probability at least $1 - \delta$,

$$\|X_n^\top W_n\|_{\max} = \max_{i,j} |[X_n^\top W_n]_{ij}| \lesssim C_{\phi\psi} \sqrt{nH \log\left(\frac{d}{\delta}\right)}.$$

*Confidence Region*

By the optimality condition, we have

$$\|Y_n - X_n(\widehat{L}_n + \widehat{S}_n)\|_F^2 \leq \|Y_n - X_n(L^* + S^*)\|_F^2.$$

Expanding on both sides yields

$$
\begin{aligned}
\|X_n(\Delta_L + \Delta_S)\|_F^2 &\leq 2\langle W_n, X_n(\Delta_L + \Delta_S)\rangle \\
&= 2\langle X_n^\top W_n, \Delta_L + \Delta_S\rangle \\
&\leq 2\|X_n^\top W_n\|_2\|\Delta_L\|_* + 2\|X_n^\top W_n\|_{\max}\|\Delta_S\|_1 \\
&\leq 2\langle X_n^\top W_n, \Delta_L + \Delta_S\rangle \\
&\leq 2\sqrt{2r}\|X_n^\top W_n\|_2\|\Delta_L\|_F + 2\sqrt{2s}\|X_n^\top W_n\|_{\max}\|\Delta_S\|_F,
\end{aligned}
$$

where we denote $\Delta_L := \widehat{L}_n - L^*$ and $\Delta_S := \widehat{S}_n - S^*$.

On the other hand, by (20) and separation lemma in (Chai & Fan, 2024), we have that

$$
\begin{aligned}
\|X_n(\Delta_L + \Delta_S)\|_F^2 &\geq c_1(n-1)\|\Delta_L + \Delta_S\|_F^2 \\
&\geq \frac{c_1(n-1)}{2}\left(\|\Delta_L\|_F^2 + \|\Delta_S\|_F^2\right).
\end{aligned}
$$

Putting together, we have

$$
\begin{aligned}
\|\Delta_L\|_F^2 + \|\Delta_S\|_F^2 &\leq \frac{4}{c_1(n-1)}\sqrt{2r}\|X_n^\top W_n\|_2\|\Delta_L\|_F + \sqrt{2s}\|X_n^\top W_n\|_{\max}\|\Delta_S\|_F \\
&\leq \frac{4}{c_1(n-1)}\sqrt{2r\|X_n^\top W_n\|_2^2 + 2s\|X_n^\top W_n\|_{\max}^2} \cdot \sqrt{\|\Delta_L\|_F^2 + \|\Delta_S\|_F^2}
\end{aligned}
$$

which implies that

$$\|\Delta_L\|_F^2 + \|\Delta_S\|_F^2 \leq \frac{32}{c_1^2(n-1)^2}\left(r\|X_n^\top W_n\|_2^2 + s\|X_n^\top W_n\|_{\max}^2\right).$$

Plugging in the aforementioned bounds of $\|X_n^\top W_n\|_2$ and $\|X_n^\top W_n\|_{\max}$, we deduce that $\mathcal{B}_n$ is a valid $\delta$-confidence region if we take

$$\beta_n = \frac{C_\beta}{n^2}\left(r(C_\phi C_\psi')^2 nH\log(d/\delta) + sC_{\phi\psi}^2 nH\log(d/\delta)\right) = \frac{C_\beta H\log(d/\delta)}{n}\left(r(C_\phi C_\psi')^2 + sC_{\phi\psi}^2\right) \tag{21}$$

for some large enough $C_\beta$. In particular, we take $\delta = 1/(N^2 H)$ in the above display, then by the union bound, $\mathbb{P}(E_N = 0) \leq N \cdot \frac{1}{N^2 H} = \frac{1}{NH}$.

Combining the definition of $\beta_n$ with (19), we obtain that

$$
\begin{aligned}
\text{Regret}(NH) &\leq C_\phi C_\psi H^2 \sum_{n=1}^{N}\sqrt{2\beta_n} + 1 \\
&\lesssim C_{reg}\sqrt{NH^5},
\end{aligned}
$$

where $C_{reg} := C_\phi C_\psi\sqrt{C_\beta\left(r(C_\phi C_\psi')^2 + sC_{\phi\psi}^2\right)\log(dNH)}$.

As a byproduct, we have that by the end of the $N$ episode, with probability at least $1 - 1/(N^2 H)$,

$$\left\|L - \widehat{L}_N\right\|_F^2 + \left\|S - \widehat{S}_N\right\|_F^2 \leq \beta_N.$$

### A.5. Discussion on Condition (20)

Now we provide an example where Condition (20) holds. At a high level, $\phi_{i,h}$ may be highly correlated, across different steps and episodes. Nonetheless, note that each episodes starts at independent initial states, hence providing diversity to the linear operator $X_n$. In fact, Condition (20) holds when $\phi(s, a)$ depends mainly on $s$ and $a$ adds a perturbation effect. To be more concrete, consider the following lemma as an example.

**Lemma A.3.** *Suppose there exists some function $\overline{\phi}$ such that $\|\phi(s, a) - \overline{\phi}(s)\|_2 \leq \eta$. And*

$$\lambda_{\min}\left(\mathbb{E}_\mu[\overline{\phi}(s)\overline{\phi}(s)^\top]\right) \geq c_{\min}.$$

*If $\eta(2C_\phi + \eta) \leq \frac{c_{\min}}{4}$ and $n \geq C_e d$ for some constant $C_e$, then with probability at least $1 - 2e^{-c_e n}$, Condition (20) holds with $c_1 = c_{\min}/4$.*

*Proof.* Denote $\mathbb{E}_\mu[\overline{\phi}(s)\overline{\phi}(s)^\top]$ by $\Sigma_\mu$. As $|\overline{\phi}(s)^\top v| \leq \|\overline{\phi}(s)\| \leq C_\phi$ for any $v \in \mathbb{S}^d$, we have that $\overline{\phi}(s)$ is subGaussian with variance proxy $(C - \phi/2)^2$. By (5.25) in (Vershynin, 2010), there exists some constants $C_e, c_e$ such that with probability at least $1 - 2e^{-c_e n}$,

$$\frac{1}{n-1}\sum_{i=1}^{n-1}\overline{\phi}(s_{i,1})\overline{\phi}(s_{i,1})^\top \succeq \frac{1}{2}\Sigma_\mu \succeq \frac{c_{\min}}{2}I$$

as long as $n \geq C_e d$.

Let $\Delta = \frac{1}{n-1}\sum_{i=1}^{n-1}\overline{\phi}(s_{i,1})\overline{\phi}(s_{i,1})^\top - \frac{1}{n-1}\sum_{i=1}^{n-1}\phi_{i,1}\phi_{i,1}^\top$, we have for any $v \in \mathbb{S}^d$ that

$$v^\top \Delta v = \frac{1}{n-1}\sum_{i=1}^{n-1}\left([\overline{\phi}(s_{i,1})^\top v]^2 - [\phi_{i,1}^\top v]^2\right)$$

$$\leq \frac{1}{n-1}\sum_{i=1}^{n-1}\left\|\overline{\phi}(s_{i,1}) - \phi_{i,1}\right\| \cdot \left\|\overline{\phi}(s_{i,1}) + \phi_{i,1}\right\|$$

$$\leq \eta(2C_\phi + \eta).$$

Hence $\|\Delta\| \leq \eta(2C_\phi + \eta) \leq \frac{c_{\min}}{4}$. It follows that

$$\frac{1}{n-1}\sum_{i=1}^{n-1}\phi_{i,1}\phi_{i,1}^\top \succeq \frac{c_{\min}}{4}I.$$

To conclude, note that

$$\frac{1}{n-1}\sum_{i=1}^{n-1}\sum_{h=1}^{H}\phi_{i,h}\phi_{i,h}^\top \succeq \frac{1}{n-1}\sum_{i=1}^{n-1}\phi_{i,1}\phi_{i,1}^\top \succeq \frac{c_{\min}}{4}I.$$

$\square$

*Remark A.4.* The condition $n \geq C_e d$ requires a warm start. For $n \leq C_e d$, one can use a fixed policy to generate samples. This will not affect the total regret as long as $dH$ is neglegible compared to $\sqrt{NH^5}$.

## B. Regret Analysis for UCB-TQL under Composite MDPs

In this section, we provide proof for Theorem 4.6. The pipeline is similar to the single-task setting.

We start by constructing the confidence region. To that end, again denote Denote $X_n^{(1)} = \begin{bmatrix} \phi_{1,1}^{(1)\top} \\ \cdots \\ \phi_{1,H}^{(1)\top} \\ \cdots \\ \phi_{n-1,H}^{(1)\top} \end{bmatrix}$ and $Y_n^{(1)} =$

$$\begin{bmatrix} \psi_{1,1}^{(1)\top} \boldsymbol{K}_\psi^{-1} \\ \cdots \\ \psi_{1,H}^{(1)\top} \boldsymbol{K}_\psi^{-1} \\ \cdots \\ \psi_{n-1,H}^{(1)\top} \boldsymbol{K}_\psi^{-1} \end{bmatrix}, \text{ we have the observational model as follows:}$$

$$Y_n^{(1)} = X_n^{(1)}(L^* + S^{*(0)} + D^*) + W_n$$

where $W_n := Y_n^{(1)} - X_n^{(1)}(L^* + S^{*(0)} + D^*)$.

By the optimality condition, we have

$$\|Y_n^{(1)} - X_n^{(1)}(\widehat{L} + \widehat{S}^{(0)} - \widehat{D}_n)\|_F^2 \le \|Y_n^{(1)} - X_n^{(1)}(L^* + S^{*(0)} + D^*)\|_F^2.$$

After some calculation, we obtain

$$\begin{aligned} \|X_n^{(1)}(\widehat{D}_n - D^*)\|_F^2 &\le 2\left\langle Y_n^{(1)} - X_n^{(1)}(\widehat{L} + \widehat{S}^{(0)} - D^*), X_n^{(1)}(\widehat{D}_n - D^*) \right\rangle \\ &= 2\left\langle X_n^{(1)\top}\left(W_n + X_n^{(1)}(L^* - \widehat{L} + S^{*(0)} - \widehat{S}^{(0)})\right), \widehat{D}_n - D^* \right\rangle \\ &\le 2\left\|X_n^{(1)\top}W_n\right\|_\infty \left\|\widehat{D}_n - D^*\right\|_1 + 2\left\|X_n^{(1)\top}X_n^{(1)}(L^* - \widehat{L} + S^{*(0)} - \widehat{S}^{(0)})\right\|_F \left\|\widehat{D}_n - D^*\right\|_F \\ &\le 2\sqrt{2e}\left\|X_n^{(1)\top}W_n\right\|_{\max} \left\|\widehat{D}_n - D^*\right\|_F + 2\left\|X_n^{(1)\top}X_n^{(1)}(L^* - \widehat{L} + S^{*(0)} - \widehat{S}^{(0)})\right\|_F \left\|\widehat{D}_n - D^*\right\|_F. \end{aligned}$$

We have, similar as before, with probability at least $1 - \delta$,

$$\left\|X_n^{(1)\top}W_n\right\|_{\max} \lesssim C_{\phi\psi}\sqrt{nH\log\left(\frac{d}{\delta}\right)}.$$

If

$$\lambda_{\min}\left(\frac{X_n^{(1)\top}X_n^{(1)}}{n-1}\right) \ge c_1,$$

$$\lambda_{\max}\left(\frac{X_n^{(1)\top}X_n^{(1)}}{n-1}\right) \le C_1,$$

then it holds that

$$\|\widehat{D}_n - D^*\|_F^2 \lesssim \frac{eC_{\phi\psi}^2 H\log\left(\frac{d}{\delta}\right)}{n} + \|L^* - \widehat{L}\|_F^2 + \|S^{*(0)} - \widehat{S}^{(0)}\|_F^2.$$

Suppose at the first stage, we established $\|L^* - \widehat{L}\|_F^2 + \|S^{*(0)} - \widehat{S}^{(0)}\|_F^2 \le \beta_{N_0}$ with probability at least $1 - 1/(2N^2H)$, where

$$\beta_{N_0} = \frac{C_\beta H\log(d/\delta)}{N_0}\left(r(C_\phi C'_\psi)^2 + sC_{\phi\psi}^2\right),$$

as in (21).

**Naive CR**   Recall that we can construct a naive confidence region as

$$\mathcal{B}_n = \left\{(L, S) \mid \left\|L - \widehat{L}_n\right\|_F^2 + \left\|S - \widehat{S}_n\right\|_F^2 \le \beta_n^{(1)}\right\}$$

where $\beta_n^{(1)} := \beta_{N_0} + \frac{eC_{\phi\psi}^2 H\log(dNH)}{n}$.

When using this confidence region to construct optimistic value functions, we can plug the value of $\beta_n^{(1)}$ into (19). It follows that

$$
\begin{aligned}
\text{Regret}(NH) &\lesssim C_\phi C_\psi H^2 \sum_{n=1}^N \sqrt{\beta_n^{(1)}} + 1 \\
&\lesssim C_\phi C_\psi H^2 \left( N \sqrt{\beta_{N_0}} + \sqrt{eC_{\phi\psi}^2 NH \log(dNH)} \right).
\end{aligned}
$$

*Remark* B.1. When $N_0 \gg N$, this bound is dominated by the second term, which depends on $C_\phi$.

**Tight CR**  As illustrated in the proof sketch, we can achieve better rate by constructing a more fine-grained confidence region as

$$
\widetilde{\mathcal{B}}_n = \left\{ (L, S, D) \mid \left\| L - \widehat{L}^{(0)} \right\|_F^2 + \left\| S - D - \widehat{S}^{(0)} \right\|_F^2 \leq \beta_{N_0}, \ \left\| D - \widehat{D}_n \right\|_F^2 \leq \beta_n^{(1)}, \ \|D\|_0 \leq e \right\}.
$$

It is straightforward to show that $(L^*, S^{*(0)}, D^*) \in \widetilde{\mathcal{B}}_n$. We then carry out a more refined one-step analysis, similar in the vein of Section 3.2.2. In particular, let $(\widetilde{L}, \widetilde{S}, \widetilde{D}) = \arg\max_{L, S \in \widetilde{\mathcal{B}}_n} \phi(s,a)^\top (L + S) \Psi^\top V_{n,h+1}$, it holds that

$$
\begin{aligned}
&Q_{n,h}(s_{n,h}, a_{n,h}) - (r(s_{n,h}, a_{n,h}) + [P_h V_{n,h+1}](s_{n,h}, a_{n,h})) \\
&\leq \left\| \phi_{n,h}^\top (\widetilde{L} - L^*) \right\|_2 \left\| \Psi^\top V_{n,h+1} \right\|_2 + \left\| \phi_{n,h}^\top \left( \widetilde{S} - \widetilde{D} - S^* + D^* \right) \right\|_2 \left\| \Psi^\top V_{n,h+1} \right\|_2 + \left| \phi_{n,h}^\top \left( \widetilde{D} - D^* \right) \Psi^\top V_{n,h+1} \right| \\
&\leq H \left( C_\psi \left\| \phi_{n,h}^\top (\widetilde{L} - L^*) \right\|_2 + C_\psi \left\| \phi_{n,h}^\top \left( \widetilde{S} - \widetilde{D} - S^* + D^* \right) \right\|_2 \right) + \sqrt{2e} H C_\phi' C_\psi \left\| \widetilde{D} - D^* \right\|_F \\
&\leq H \left( C_\psi \|\phi_{n,h}\|_2 \left\| \widetilde{L} - L^* \right\|_F + C_\psi \|\phi_{n,h}\|_2 \left\| \widetilde{S} - \widetilde{D} - S^* + D^* \right\|_F \right) + \sqrt{2e} H C_\phi' C_\psi \left\| \widetilde{D} - D^* \right\|_F \\
&\leq C_\phi C_\psi H \left( \left\| \widetilde{L} - L^* \right\|_F + \left\| \widetilde{S} - \widetilde{D} - S^* + D^* \right\|_F \right) + \sqrt{2e} H C_\phi' C_\psi \left\| \widetilde{D} - D^* \right\|_F \\
&\leq C_\phi C_\psi H \sqrt{2\beta_{N_0}} + C_\phi' C_\psi H \sqrt{4\beta_n^{(1)} e},
\end{aligned}
\tag{22}
$$

where the sparsity constraint on $D$ is used in bound the third term in the second line. Combined with the one-step error in the regret decomposition (19), we obtain that

$$
\begin{aligned}
\text{Regret}(NH) &\lesssim C_\phi C_\psi H^2 \sum_{n=1}^N \sqrt{\beta_{N_0}} + C_\phi' C_\psi H^2 \sum_{n=1}^N \sqrt{4\beta_n^{(1)} e} \\
&\lesssim \left( C_\phi + C_\phi' \sqrt{e} \right) C_\psi H^2 N \sqrt{\beta_{N_0}} + C_\phi' C_\psi H^2 \sqrt{eC_{\phi\psi}^2 NH \log(dNH)}.
\end{aligned}
$$

Plugging the definition of $\beta_{N_0}$ completes the proof.

*Remark* B.2. When $N_0 \gg N$, this bound is dominated by the second term, which does not depend on $C_\phi$, but rather on $C_\phi'$. It is tighter than the rate of naive CR approach as $C_\phi \geq C_\phi'$.

