# OpenReview forum: "Transfer Q-Learning with Composite MDP Structures"
_ICML.cc/2025/Conference — ICML 2025 poster_

### Official Review · Reviewer_CS8V · 2025-02-27

**Overall Recommendation:** 2

**Summary:**

The paper addresses transfer reinforcement learning (RL) under a new framework model called composite MDP, where transition dynamics consist of a low-rank “shared” component plus a sparse “task-specific” component. This setup reflects how different tasks can share a core set of dynamics while still varying in a limited number of ways.

The authors first studies single-task Learning with composite MDPs. They present a UCB-based algorithm (UCB-Q-Learning) and provide regret bounds that extend standard results in linear and low-rank MDPs to this more flexible composite structure.

Then the authors consider transfer learning across tasks, where in the source and target tasks, the low-rank parts are shared, and only the sparse components differ. They introduce a transfer Q-learning algorithm (UCB-TQL) to leverage a previously learned source-task model. They prove that when the difference between the target task and source tasks' sparse components is very small, the regret bound for the target task becomes nearly dimension-free w.r.t. ambient dimension.

**Claims And Evidence:**

The term **“low-rank”** in this paper appears to be used inconsistently, leading to ambiguity.

In summary, I find three different interpretations of “low-rank” within the paper:

1. **"Low-rank MDPs"**
   - In my understanding, the notion of "low-rank" in **low-rank MDPs** compares the embedding dimension $d$ with the original large state space. As long as $ d \ll |\mathcal{S}| $, the MDP should still be considered low-rank.

2. **"Transition core** $M^*$ **is no longer a low-rank matrix"**  (Line 53, Right Column)
   - Here, "not low-rank" likely refers to the high-dimensional setting, where $ p, q \gg N $.


3. **"Where** $L^* $  **is a low-rank incoherent matrix"**  (Line 163, Right Column)
   - I did not find any further explanation of the **low-rank structure** of $L^*$, so its exact role seems unclear. Also I wonder whether this structure contributes to solving the high-dimensional problem.

I find these differing interpretations make the discussion difficult to follow. For instance, the paper states:

> *"Similarly, methods built for low-rank MDPs fail in our context due to the absence of low-rank assumptions in $M^* $."* (Line 57)

However, even if $M^* $ does not satisfy a low-rank assumption, the model may still be a low-rank MDP in the sense of an embedding-based formulation.  So why do methods for low-rank MDPs fail if the overall structure is still low-rank?

I think clarifying these distinctions would improve the paper's readability and consistency.

**Essential References Not Discussed:**

The composite MDPs also seem to be a special case of a linear factored MDPs. [1,2]. I think there should be some discussions on how these models are related.

[1] Sample-Optimal Parametric Q-Learning Using Linearly Additive Features.

[2] Model-Based Reinforcement Learning with Value-Targeted Regression.

**Experimental Designs Or Analyses:**

There are no experiments.

**Methods And Evaluation Criteria:**

The algorithm is standard UCB-type algorithm and the evaluation criteria is standard regret.

**Other Comments Or Suggestions:**

Please add the Impact Statement section.

**Other Strengths And Weaknesses:**

Weakness:

1. Computational Efficiency of  $K_\Psi$  in Large State Spaces. When the state space $\mathcal{S}$  is infinite or extremely large (which should be considered given the assumed high-dimensional setting), how can  $K_\Psi$  be computed efficiently, even if  $\Psi$  is known?
Previous works, such as [1], typically store only the covariance matrix of the sampled features, making their approach computationally efficient. How does your method compare in terms of efficiency?

Question:

1. Clarification on Theorem 3.6. Could you elaborate more on Theorem 3.6? In Remark 3.7, you mention that the result matches previous bounds in terms of  $d$ , which have polynomial dependence on $d$, but your bound does not explicitly involve any polynomial dependence on  $d$. Do  $r$  and  $s$  scale as poly$( d )$? This is crucial because you assume  $p, q \gg N$; if the upper bound includes  $O(d)$ , the regret will scale superlinear with $N$. If the result does not involve polynomial dependence on  $d$, could you provide a high-level explanation of why your method can handle the high-dimensional setting, whereas low-rank and linear MDP methods fail?

[1] Provably Efficient Reinforcement Learning with Linear Function Approximation.

**Questions For Authors:**

Please see Other Strengths And Weaknesses

**Relation To Broader Scientific Literature:**

The proposed model composite MDPs is different from previous literature, but more discusssions on the relationship with previous literature are required. Please see below and "Claims And Evidence".

**Theoretical Claims:**

I go through the proof of theorem 3.6, which seems correct.

---

> ### Author Rebuttal · Authors · 2025-03-31
>
> Thank you for your insightful comments!
>
> **Low-Rank:** We agree that clarifying this terminology is important for improving readability and avoiding confusion. In classical low-rank MDPs literature, "low-rank" refers to embedding-based models where the effective feature dimension $d$ is small compared to $|\mathcal{S}|$. In contrast, in our setting, we work in a high-dimensional regime with $p, q \gg N$, so $M^*$ is not globally low-rank.  Prior methods developed for low-rank MDPs—which assume a small feature dimension—do not apply directly, since they cannot consistently estimate $M^\ast$ within $N$ trajectories.
>
> To resolve this issue, we assume $M^* = L^* + S^*$, where $L^*$ is of low-rank $r$ and $S^*$ is sparse. This structure enables consistent estimation in high dimensions and is key to our theoretical analysis. Crucially, this assumption enables us to design an estimator that achieves *minimax-optimal error rates* for recovering $L^\ast$, $S^\ast$, and consequently $M^\ast$, *independently of the ambient dimension* $d = \max (p, q)$.
>
> We have revised the manuscript to clarify the distinction between embedding-based low-rank MDPs and our structured high-dimensional setting.
>
> **Prior Work [1,2]:** Our model can be written in the form of [1] as $P(s' \mid s, a) = \phi(s,a)^\top \alpha(s')$ with $\alpha(s') := (L^* + S^*)\psi(s')$. This reduces to [1] when $S^* = 0$. Similarly, our model matches the linear factored MDP in [2] via a Kronecker formulation: $P(s' \mid s,a) = (\phi(s,a) \otimes \psi(s'))^\top \operatorname{vec}(L^* + S^*)$.
> While structurally related, a key difference lies in estimation. Our setting allows $p, q \gg N$, and our estimator achieves minimax-optimal error rates that do **not** scale with $d = \max(p, q)$. In contrast, [1,2] assume low-dimensional or identifiable parameter spaces and are not suitable for high-dimensional regimes. Our structured assumption is especially crucial in **transfer RL**, enabling effective knowledge sharing via $L^*$ while capturing task-specific deviations via $S^*$.
>
> **Comp. Efficiency of $K_\psi$:** We agree this is important. When $|\mathcal{S}|$ is large or infinite, we compute $K_\psi$ using a Monte Carlo approximation:
> $
> \hat{K}\_\psi = \frac{1}{m} \sum\_{i=1}^m \psi(s\_i)\psi(s\_i)^\top, \quad s\_i \sim \text{Unif}(\mathcal{S}).
> $
> This is a standard approach in randomized numerical linear algebra [3] and is computed only **once** before online learning. Our method is thus comparable in efficiency to [1], which stores empirical covariances, but we use $K_\psi$ to build confidence regions specific to our model.
>
> **Clarification on Theorem 3.6:** Our method handles the high-dimensional setting by explicitly exploiting the low-rank-plus-sparse structure of the transition core matrix $M^* = L^* + S^*$, where $L^*$ is of low-rank $r << d$ and $S^*$ is of sparsity $s << d$. This structure enables consistent estimation in regimes where $p, q \gg N$, and is key to our theoretical guarantees. We have updated Remark 3.7 to clearly state the point.
>
> Crucially, our estimation procedure achieves *minimax-optimal error rates* for $L^*$, $S^*$, and $M^*$ that are **independent of the ambient dimension** $d = \max(p, q)$. We do **not** assume that the rank $r$ or sparsity level $s$ scale with $d$; under Assumption 3.1, both can remain small (e.g., constant or logarithmic in $d$), ensuring the estimation error bound does not grow with $d$ even in high-dimensional settings and our transfer learning result is **explicitly free of $d$**, demonstrating that the benefit of structural transfer carries over without dependence on the ambient dimension.
>
> **Technical Note and Transfer RL:** Our regret analysis relies on bounding Frobenius norm errors and carefully transferring them to Bellman recursion. Improvements using variance reduction or stronger structural assumptions are possible future work. A major contribution of our paper is in **transfer RL**: our low-rank-plus-sparse structure enables sample-efficient transfer, capturing shared structure via $L^*$ and task-specific deviations via $S^*$. Prior work like FLAMBE [Agarwal et al., 2020] does not handle this generality. We have added clarifications throughout the revised paper.
>
> We thank the reviewer again for your support and helpful comments, and we hope the improvements further affirm the strength and clarity of our submission.
>
> **References:**
>
> [1] Sample-Optimal Parametric Q-Learning Using Linearly Additive Features
> [2] Model-Based Reinforcement Learning with Value-Targeted Regression
> [3] Drineas & Mahoney, RandNLA: randomized numerical linear algebra, CACM 2016

---

### Official Review · Reviewer_fH8W · 2025-03-12

**Overall Recommendation:** 3

**Summary:**

This paper introduces a composite MDP framework combining low-rank shared dynamics and sparse task-specific variations, along with the UCB-TQL algorithm for transfer Q-learning. Theoretically, it establishes a dimension-free regret bound for the target task by leveraging structural similarities between tasks. While the work provides rigorous theoretical guarantees, empirical validation and comparisons with existing methods remain open questions.

**Claims And Evidence:**

Yes, the claims are generally supported by clear and convincing evidence.

**Essential References Not Discussed:**

No, the essential references are discussed in the paper.

**Experimental Designs Or Analyses:**

This paper primarily focuses on theoretical analysis and lacks experimental validation.

**Methods And Evaluation Criteria:**

Yes. The methods are largely appropriate for addressing the challenges of transfer reinforcement learning.

**Other Comments Or Suggestions:**

Page 3, Definition 2.1: "probability transition model $P: \mathcal{S} \times \mathcal{A} \rightarrow \bigtriangleup(\mathcal{A})$" -> Should clarify $ \bigtriangleup(\mathcal{S})$ instead of $\bigtriangleup(\mathcal{A})$.

**Other Strengths And Weaknesses:**

Strengths:
- This paper introduces a low-rank + sparse composite MDP to model shared dynamics and task-specific variations, addressing limitations of prior work that assumes purely low-rank structures.
- It provides a very detailed theoretical analysis and proof.

Weaknesses:
- While theoretically sound, the paper does not include experiments to validate UCB-TQL’s performance on benchmarks or compare it with existing transfer RL methods, leaving practical efficacy unverified.
- Assumes tasks differ only along sparse dimensions, ignoring more complex inter-task relationships. Real-world tasks often exhibit more complex deviations, limiting the framework’s generality.

**Questions For Authors:**

Please see Weaknesses and Suggestions.

**Relation To Broader Scientific Literature:**

The paper introduces a composite MDP framework combining low-rank shared dynamics with sparse task-specific components, and proposes UCB-TQL for transfer RL with dimension-independent regret bounds. Theoretical analysis demonstrates that UCB-TQL effectively exploits structural similarities while adapting to task variations, achieving improved sample efficiency over single-task RL through rigorous confidence region construction for sparse differences.

**Theoretical Claims:**

Yes. The theoretical claims are sound.

---

> ### Author Rebuttal · Authors · 2025-03-31
>
> **Response to Reviewer fH8W**
>
> We sincerely thank the reviewer for the thoughtful feedback and recognition of our theoretical contributions. We address the two concerns raised below.
>
> ---
>
> **Q1: Lack of empirical validation of UCB-TQL**
>
> We appreciate the reviewer’s point. While this submission does not include empirical evaluations, our goal is to provide a *theoretical foundation* for transfer reinforcement learning (RL) in high-dimensional settings with structured heterogeneity. Specifically, UCB-TQL is the **first algorithm with provable regret guarantees** under the composite MDP model that includes *both low-rank and sparse deviations*, and achieves $\widetilde{\mathcal{O}}(\sqrt{e} H^5 N)$ regret that is **independent of ambient dimension**.
>
> This setting is motivated by real-world applications (e.g., autonomous driving, robotics, and healthcare), where tasks often share underlying dynamics but differ in subtle, structured ways. In such domains, the transition dynamics may be high-dimensional, and assuming *pure low-rank* or *dense similarity* across tasks can be too simplistic. The low-rank-plus-sparse decomposition we propose reflects this nuanced task structure and provides an *interpretable and tractable* way to model variation.
>
> Although we do not present experiments here, we see our work as a **theoretical counterpart** to recent empirical transfer RL methods. Our regret guarantees offer insights into *when and how transfer should help*, which is often unclear in empirical work. We believe our analysis will inspire future algorithms that combine empirical effectiveness with theoretical robustness, and we are currently implementing UCB-TQL in benchmark settings.
>
> We hope the community will view our contribution as laying the groundwork for **provably efficient transfer in high-dimensional RL**, and we are **excited to follow up** with empirical validation in future work.
>
> ---
>
> **Q2: Limitation of sparsity-based task differences**
>
> We fully agree that real-world tasks can exhibit more complex deviations than those modeled by sparse differences. Our current work makes a *structured but generalizable starting assumption*: tasks differ sparsely in transition dynamics after aligning shared core structure. This assumption enables us to design statistically and computationally efficient estimators, while still covering rich task classes.
>
> We appreciate the suggestion and have added discussion in the revised manuscript exploring possible **extensions**:
>
> - **Task-specific low-rank components**: One natural direction is to allow each task to have its own low-rank variation on top of a shared component. This introduces additional complexity but could capture broader relationships.
> - **Decomposable low-rank spaces**: Another extension is to learn a union of low-rank subspaces across tasks, leveraging techniques from subspace clustering and matrix factorization.
> - **Feature selection on $\Phi$ and $\Psi$**: When feature sets are known but high-dimensional, incorporating feature selection or structured sparsity could adaptively identify task-relevant components.
> - **Group-sparse or structured differences**: Beyond elementwise sparsity, our model could be extended to handle block or group-structured variations, which would better capture correlated shifts.
>
> We view our work as a first step toward unifying high-dimensional structure with transfer learning, and we are enthusiastic about these extensions.
>
> ---
>
> **Closing Remarks**
>
> We again thank the reviewer for the constructive comments. We believe our theoretical framework makes a timely and valuable contribution to the transfer RL literature, particularly in advancing the understanding of *when and how transfer helps* in high-dimensional environments. By addressing the above concerns, we hope to improve the clarity, flexibility, and practical relevance of our work. We respectfully hope the reviewer may consider increasing their score.

---

### Official Review · Reviewer_ZhAb · 2025-03-14

**Overall Recommendation:** 4

**Summary:**

The paper introduces a Upper Confidence Bound method for Transfer Q-Learning for transfer RL settings where it is assumed that transition dynamics are assumed to decompose to a low-rank shared matrix and a sparse matrix that captures task specific dynamics. A key feature is that the method allows for high-dimensional transition dynamics. The paper follows on to provide theoretical guarantees on the UCB Q-learning applied to this structure for single-task learning as well as transfer. Transfer occurs by fixing the low-rank component and adapting the sparse component.

**Claims And Evidence:**

I believe that the key claims of the paper are verified through the theorems up to their correctness.

**Essential References Not Discussed:**

None to my knowledge.

**Experimental Designs Or Analyses:**

This is a theoretical paper; no experiments were reported.

**Methods And Evaluation Criteria:**

This is a theoretical paper; no experiments were reported.

**Other Comments Or Suggestions:**

### Section 2
- Could the authors be more explicit about the form of $P$. Is it a function $\mathcal{S} \times \mathcal{A} \times \mathcal{S} \rightarrow [0, 1]$? If so, I am not sure I understand the notation $[PV_{h+1}^\pi](s, a)$.

### Section 3
- Assumption 3.1(i) - Please specify what $\mu$ and $r$ are.
- Equation 3 - Please state that $|| \cdot ||_F$ refers to the Frobenius norm (which I assume is what it means).

**Other Strengths And Weaknesses:**

### Strengths
- Good literature review
- Other than the weakness mentioned below, it is written well.
### Weaknesses
- The notation in some places can be better explained.
- There are no examples of experiments run with the proposed method to show how well the theory can be utilized in practice.

**Questions For Authors:**

1. line 154: What does the notation $[H]$ mean. I think it means the range from $[0, H]$. If so, why not say that, or specify what the notation means.
2. line 191: Is the summation in $\mathbf{K}_\psi = \sum_{s'\in \mathcal{S} ...$ a mistake? I don't think the simplification in step 2 of Equation (1) works with this sum.
3. line 210: What does the notation $|| \cdot ||_{2, \infty}$ mean? Specifically, does it mean the 2-norm and $\infty$-norm, or all norms from 2 to $\infty$, or something else?
4. Could you give examples of where the specific 'low rank core transition with sparse task specific structures' arise. Could you also give counter examples?

**Relation To Broader Scientific Literature:**

The paper builds on top of Composite MDPs and Q-learning with UCB.

**Theoretical Claims:**

The paper provides several theorems. While I have gone through the proofs, and believe that they are correct, I would not say that I have verified them rigorously.

---

> ### Author Rebuttal · Authors · 2025-04-01
>
> **We thank the reviewer for their positive assessment and constructive suggestions.** We are especially grateful for the recognition of our theoretical contribution, which we believe offers a timely and foundational advancement in transfer reinforcement learning.
>
> Our work provides the *first provable regret guarantees* for transfer RL in high-dimensional settings where the transition dynamics exhibit structured heterogeneity. By modeling transitions as $M^* = L^* + S^*$—with $L^*$ low-rank and $S^*$ sparse—we capture realistic task variations (e.g., shared core dynamics with sparse task-specific shifts). This structure allows us to derive *minimax-optimal error rates* for estimating $L^*$ and $S^*$, and achieve regret bounds that are *independent of the ambient dimension* $d = \max(p, q)$. These insights lay a theoretical foundation that complements ongoing empirical advances and help answer the fundamental question: *when and how does transfer learning help in high-dimensional RL?*
>
> We have revised the manuscript to improve clarity in notation and address specific concerns raised:
>
> - **Section 2 – Notation of $\mathcal{P}$ and $\mathcal{P}V^\pi_{h+1}(s,a)$:**
>   $\mathcal{P}$ is a linear operator mapping a function $V : \mathcal{S} \to \mathbb{R}$ to a function over $(s,a)$ via $\mathcal{P}V^\pi\_{h+1}(s,a) = \sum\_{s'} \mathcal{P}(s'|s,a) V^\pi\_{h+1}(s')$. We have clarified this in the updated text.
>
> - **Line 154 – $[H]$ Notation:**
>   Yes, you are right. $[H]$ denotes the index set $\{1,2,\dots,H\}$. We have clarified this in the updated text.
>
> - **Line 191 – Summation in $K_\psi$ and Equation (1):**
>   The equality in Equation (1) is valid. $K_\psi$ is defined as the sum over all $s' \in \mathcal{S}$ and is independent from the outer summation over $s' \in \mathcal{S}$ in Equation (1). The simplification extracts common factors based on our composite MDP formulation.
>
> - **Line 210 – $\|\cdot\|_{2,\infty}$ Notation:**
>   This denotes the 2-to-infinity operator norm, i.e., the maximum $\ell_2$ norm of the rows. We've added this to the notation section for clarity.
>
> - **Equation (3) – Frobenius Norm:**
>  $\|\cdot\|_F$ refers to the Frobenius norm and was defined in the supplementary material, now clarified in the main text.
>
> - **Assumption 3.1 – Meaning of $\mu$ and $r$:**
>   Thank you for your careful reading! $\mu$ is the incoherence parameter (a constant > 1), and $r$ is the rank of the low-rank component $L^*$, now clarified in the main text.
>
> Regarding examples of our model, consider **personalized healthcare**: the low-rank component captures typical patient responses to treatments, while the sparse component models anomalies or individual-specific deviations. Other real-world domains include **recommendation systems** and **robotics**, where core dynamics are shared but task-specific noise or variation exists. A **counterexample** would be settings where task variation arises from structured transformations or complex nonlinear mappings—scenarios that go beyond additive sparse deviations and are natural directions for future work.
>
> We thank the reviewer again for their support and helpful comments, and we hope the improvements further affirm the strength and clarity of our submission.

---

> > ### Comment · Reviewer_ZhAb · 2025-04-09
> >
> > Thank you for the clarifications.
> >
> > Regarding \textbf{line 191}, my understanding of your rebuttal is that the index $s'$ in line 191 is different to the index $s'$ in Equation (1). Wouldn't using a different notation for this make sense, or do you think it is clear from context?
> >
> > I will leave my score as is since I think my concerns other than the above have been addressed.

---

### Decision · Program_Chairs · 2025-05-01

**Decision:**

Accept (poster)

**Comment:**

This paper proposes a novel theoretical framework, the composite MDP (low-rank shared + sparse task-specific dynamics), and associated UCB-based algorithms (UCB-Q, UCB-TQL) for single-task and transfer reinforcement learning in high dimensions. Reviewers acknowledged the novelty of the composite model and the significance of the theoretical contributions, particularly the derived regret bounds that achieve near dimension-free dependence on the ambient dimension under specific transfer conditions. The detailed theoretical analysis was also noted as a strength. However, a major and unanimous criticism is the complete absence of any experimental validation. This lack of empirical results prevents assessment of the practical performance, applicability, or comparison of the proposed methods against existing approaches. Significant concerns were also raised regarding clarity and consistency, especially the ambiguous use of the term "low-rank," alongside requests for clarification on specific notations and theorem details. Furthermore, questions about the computational efficiency of the proposed algorithms in large state spaces and the generality of the structural assumptions relative to real-world task variations were highlighted. While the theoretical advancements are recognized, the lack of experiments and clarity issues significantly detract from the work, leading to a mixed assessment leaning slightly negative.